# Dual mTOR/DNA-PK Inhibitor CC-115 Induces Cell Death in Melanoma Cells and Has Radiosensitizing Potential

**DOI:** 10.3390/ijms21239321

**Published:** 2020-12-07

**Authors:** Felix Bürkel, Tina Jost, Markus Hecht, Lucie Heinzerling, Rainer Fietkau, Luitpold Distel

**Affiliations:** 1Department of Radiation Oncology, Universitätsklinikum Erlangen, Friedrich-Alexander-Universität Erlangen-Nürnberg, Universitätsstr. 27, 91054 Erlangen, Germany; felix.buerkel@fau.de (F.B.); tina.jost@uk-erlangen.de (T.J.); markus.hecht@uk-erlangen.de (M.H.); rainer.fietkau@uk-erlangen.de (R.F.); 2Department of Dermatology, Universitätsklinikum Erlangen, Friedrich-Alexander-Universität Erlangen-Nürnberg, Ulmenweg 18, 91054 Erlangen, Germany; lucie.heinzerling@uk-erlangen.de

**Keywords:** DNA-PK, mTOR, melanoma, DNA repair, radiosensitivity, non-homologous end-joining (NHEJ), homologous recombination (HR), kinase inhibitor

## Abstract

CC-115 is a dual inhibitor of the mechanistic target of rapamycin (mTOR) kinase and the DNA-dependent protein kinase (DNA-PK) that is currently being studied in phase I/II clinical trials. DNA-PK is essential for the repair of DNA-double strand breaks (DSB). Radiotherapy is frequently used in the palliative treatment of metastatic melanoma patients and induces DSBs. Melanoma cell lines and healthy-donor skin fibroblast cell lines were treated with CC-115 and ionizing irradiation (IR). Apoptosis, necrosis, and cell cycle distribution were analyzed. Colony forming assays were conducted to study radiosensitizing effects. Immunofluorescence microscopy was performed to determine the activity of homologous recombination (HR). In most of the malign cell lines, an increasing concentration of CC-115 resulted in increased cell death. Furthermore, strong cytotoxic effects were only observed in malignant cell lines. Regarding clonogenicity, all cell lines displayed decreased survival fractions during combined inhibitor and IR treatment and supra-additive effects of the combination were observable in 5 out of 9 melanoma cell lines. CC-115 showed radiosensitizing potential in 7 out of 9 melanoma cell lines, but not in healthy skin fibroblasts. Based on our data CC-115 treatment could be a promising approach for patients with metastatic melanoma, particularly in the combination with radiotherapy.

## 1. Introduction

The stability and integrity of the DNA is substantial for its function. Genomic instability plays a major role in cancer development and is often based on mutations in DNA repair genes [1]. Thus, taking advantage of the thereby obtained increased sensitivity to DNA-damaging agents may enhance effectiveness of cancer therapeutics [2,3,4]. Targeting the DNA-damage response (DDR) is evidenced by numerous efforts to induce selective cell death [5,6]. Since radiation is known to cause DNA double-strand breaks (DSB) and subsequent cell death [7,8], radiotherapy is a key component of cancer therapy and is received by approximately 50% of all cancer patients [9,10]. DSBs are difficult to repair and rank among the most deleterious forms of DNA damage [11]. In general, there are two main DSB-repair pathways: homologous recombination (HR) and non-homologous end-joining (NHEJ) [12]. HR is an error-free repair system occurring only when undamaged sister-templates are available [13]. In contrast, NHEJ, a more common process, can directly mediate the re-ligation of broken DNA molecules [14,15,16].

The DNA-dependent protein kinase (DNA-PK), a serine/threonine kinase, has been implicated in a variety of processes such as transcription, the regulation of gene expression and the activation of innate immunity, but its primary role is to initiate NHEJ [11,17,18,19]. DNA-PK consists of PRKDC (DNA-PKcs), a large catalytic subunit, and Ku, a heterodimeric protein [20]. The Ku heterodimer recognizes and binds to the free ends of DSBs as a first step of NHEJ and recruits the canonical factors involved in NHEJ, including PRKDC [21,22]. PRKDC operates as a scaffold protein that facilitates the loading of repair proteins to the site of damage and phosphorylates numerous substrates through its catalytic activity which promotes the execution of the DNA damage repair [23]. Different inhibitors of DNA-PK have been developed recently [24,25] and are currently evaluated in clinical trials for patients with various malignancies (e.g., NCT02316197, NCT03770689).

The mechanistic (formerly “mammalian”) target of rapamycin (mTOR) is a serine/threonine kinase interacting with diverse proteins to form two distinct complexes named mTOR complex 1 (mTORC1) and 2 (mTORC2) [26]. The mTOR protein has a crucial role in cell proliferation, metabolism, autophagy and survival [27,28]. The upregulation of mTOR signaling has tumor-inducing potential by promoting its growth and progression and the aberrant regulation of mTOR has been observed in a variety of cancers [27,29]. Indeed, mTOR inhibitors have been approved for the cancer treatment of renal cell carcinoma and mantle cell lymphoma and are in clinical trial for other indications [30,31,32,33].

CC-115 is a dual mTORC1/mTORC2 and DNA-PK inhibitor that is currently analyzed in clinical trials [34,35,36]. The anti-tumor effects of CC-115 in vitro have been shown in hepatocellular, breast, head, and neck, as well as hematological and non-small-cell lung cancer [34,35]. Currently, CC-115 is the subject of a phase Ia/b study in patients with glioblastoma multiforme (GBM), castration-resistant prostate cancer, head and neck squamous carcinoma, chronic lymphocytic leukemia, small lymphocytic lymphoma, and Ewing sarcoma (NCT01353625) [37] and a phase Ib study in men with castration-resistant prostate cancer (NCT02833883). Furthermore, a phase II study investigating the combination of CC-115 treatment and radiotherapy in patients with GBM is currently in progress (NCT02977780). Tumor biopsies from CC-115-treated GBM patients showed that CC-115 passes the blood-brain barrier [37].

Melanoma is a type of skin cancer that originates from melanocytes [38]. The incidence of melanoma is rising faster than that of any other solid tumor [39,40]. Among all malignancies, melanoma has the highest risk to spread to the brain besides lung cancer [41]. Notably, brain metastases represent the main cause of death in melanoma patients [42]. Radiotherapy is frequently used to treat metastases, e.g., in the brain with stereotactic radiotherapy or bone with fraction regimens of 2 Gy up to a dose of 50 Gy [43,44,45,46,47]. Also, single symptomatic metastases can be irradiated if systemic treatment controls disease in the rest of the body. In recent years, targeted therapies such as BRAFMEK inhibitor therapy and immune check point inhibitors with anti-CTLA4/PD-1 antibodies have improved the overall survival of patients with metastatic melanoma [48,49] and the combination of radiotherapy with systemic therapies has been thoroughly evaluated [50,51,52,53]. Nevertheless, outcomes are still dissatisfying, and improvement is required. Furthermore, there are many patients who cannot be treated sufficiently with the existing therapies. Therefore, there is a need to identify new approaches that provide alternatives to or complement the existing treatments.

In this study, we examined the effects of CC-115 on melanoma cells in comparison to healthy-donor skin fibroblasts. Fibroblasts were used because they are the main cause of undesirable side effects such as fibrosis. Our research objective was to determine whether CC-115 increases cell death with a possible radiosensitizing effect. To assess the differences between malignant and non-malignant cell lines and potential underlying molecular mechanisms, we carried out flow cytometry for apoptosis/necrosis as well as cell cycle analysis. Colony forming assays were performed to evaluate possible radiosensitizing effects of CC-115 and immunofluorescence microscopy was performed to determine HR activity.

## 2. Results

We studied the effect of dual mTOR and DNA-PK inhibitor CC-115 (Figure 1A) in combination with IR. CC-115 can inhibit NHEJ and therefore impair the repair of IR induced DSBs. We used two skin fibroblast cell cultures of healthy donors, three melanoma cell lines, and six patient-derived melanoma cell cultures. In a dose-escalation study (Figure 1B) cell death increased linearly with dose of CC-115 in skin fibroblasts (SBLF7) and reached saturation in the clearly more sensitive melanoma cells (ARPA, HV18MK). The displayed half maximal inhibitory concentrations (IC50) for the different cell lines (Figure 1B) underline that healthy-donor skin fibroblasts are less sensitive. Considering, that the combination of inhibitor treatment and IR could increase cell death rates vigorously, for further analysis we chose concentrations of CC-115 that resulted in moderate cell death rates during the dose-escalation study. Additionally, chosen concentrations were supposed to be in line with concentrations used in previous pre-clinical studies. Therefore, concentrations of 2 and 5 µmol/L CC-115 were chosen for the following experiments.

### 2.1. Apoptosis and Necrosis Analysis

Cells were treated with 2 and 5 µmol/L CC-115, IR alone, and with the combination of inhibitor and IR. Dead cells were defined as the sum of apoptotic and necrotic cells detected using flow cytometry after Annexin V-APC and 7-AAD staining. Unstained cells (Annexin-negative/7-AAD-negative) were defined as viable cells. Annexin-positive/7-AAD-negative cells were defined as apoptotic and Annexin-positive/7-AAD-positive cells as necrotic. Representative dot plots of SBLF9, a skin fibroblast cell culture and Mel624, a melanoma cell line, are depicted (Figure 2A). The proportion of dead cells with increasing inhibitor concentration in the presence and without 2 Gy IR is shown (Figure 2B).

After two days of incubation, effects of IR alone compared to control tended to be rather small or non-existent. In nine out of eleven cell lines/cultures an increasing concentration of CC-115 led to increased cell death (*p* = 0.05). In contrast, we observed no or only small differences regarding SBLF9 and ANST. SBLF7, PMelL and A375M showed a small and LIWE and ICNI a medium increase of cell death at different drug concentrations with and without IR. We noticed strong effects of CC-115 treatment alone and in combination with IR in RERO, ARPA, H18MK, and especially Mel624 cells.

### 2.2. Cell Cycle Analysis

The mTOR protein plays an important role in cell cycle regulation, especially for the G1/S transition [55]. Since CC-115 not only inhibits DNA-PK but also mTOR, cell cycle analysis was performed to determine possible inhibitory effects on the cell cycle. Depending on the cell cycle phase, cells react differently to radiation. Cells are most sensitive to radiation during mitosis and G2 phase [56]. Considering this, cell cycle analysis is coherent. The resulting histograms of SBLF9 and Mel624 are depicted (Figure 3A). Appendix A shows the gating strategy and the method of excluding doublets more detailed. The distribution of cells n G0/G1 phase and G2M phase is displayed below (Figure 3B).

Following inhibitor treatment, we observed a significant increase in the proportion of cells in G0/G1 phase (*p* = 0.05) in comparison to control in five out of nine melanoma cell lines/cultures including Mel624, HV18MK, ARPA, ICNI, and PMelL (Figure 3B). Except for ICNI, these cell lines also showed a significant decrease in the proportion of cells in the G2/M phase (*p* = 0.05). The proportion of cells in G0/G1 phase did not differ between control and treatment groups or in some cases we observed a decrease in the other melanoma cell lines/cultures, namely LIWE, A375M, ANST, and RERO, as well as in skin fibroblast cell cultures SBLF9 and SBLF7 (Figure 3B).

### 2.3. Colony Forming Assay

The colony formation assay was chosen to determine cell reproductive death and radiosensitization after IR treatment. The assay is also used to evaluate the effectiveness of drug-induced cytotoxicity [57,58]. Therefore, we analyzed the cells following CC-115 treatment alone, IR alone, and the combination of both by using colony formation assay. We chose the melanoma cell culture HV18MK to show inhibitor-dose dependent survival using up to 5 µmol/L CC-115 (Figure 4A) and also to investigate possible differences in cell survival between treated (1 µmol/L CC-115) and untreated cells while increasing IR up to 5 Gy (Figure 4B). For the more sensitive colony formation assay, lower CC-115 concentrations had to be used.

Increasing CC-115 concentration led to a decrease in cell survival (Figure 4A). Significantly less HV18MK cells treated with 1 µmol/L CC-115 survived at any IR dose compared to untreated cells (*p* = 0.016, Figure 4B). This is also observed for the normalized 1 µmol/L CC-115 curve compared to the non-treated curve (*p* = 0.031, Figure 4B). For further colony forming assays besides control, we chose three conditions: 2 Gy IR, 1 µmol/L CC-115 and 1 µmol/L CC-115 + 2 Gy IR. We chose identical cell cultures/lines for colony forming assays as for flow cytometry analysis. It was not possible to perform the assay with melanoma cell cultures ICNI and ANST because these cells did not grow as single cells. The survival fraction (SF) following CC-115 exposure alone was distinctly lower than control in all cell lines except for RERO (Figure 4C). Combined treatment clearly led to lowest SFs compared to CC-115 treatment alone or IR alone in any cell culture (Figure 4C). Dose enhancement factors (DEF) were calculated for each cell line as ratio of the isoeffective radiation dose at survival of 70% of untreated cells to the normalized CC-115-treated cells. We chose a isoeffective survival rate of 70% because the highest survival in the normalized 1 µmol/L CC-115 curve at 2 Gy was 64%. For the calculation of DEF we therefore could not fall short of this percentage. DEF > 1 indicates that the inhibitor treatment induced radiosensitivity [24,59,60,61]. Normalized CC-115 treatment showed clearly increased effects for combination of inhibitor and IR in RERO (DEF_0.7_ = 2.6), LIWE (DEF_0.7_ = 1.5), ARPA (DEF_0.7_ = 1.4), Mel624 (DEF_0.7_ = 1.4) and HV18MK (DEF_0.7_ = 1.9). No additive effects (DEF_0.7_ = 1.0) were found in SBLF9, SBLF7, PMelL, and A375M. The lowest SF was observed during combined IR and inhibitor treatment in HV18MK.

### 2.4. Homologous Recombination Assay

When blocking the second major DDR pathway NHEJ, the cells should be forced to use HR. RAD51 plays a central role in the HR [62,63]. If cells are able to use HR proficiently the amount of RAD51 foci should increase, whereas cells with deficiency in HR should show a stable or decreasing number of foci. In our assay, cells were forced to use HR by blocking NHEJ via the DNA-PK inhibitor CC-115. A dose of 10 Gy induced DSBs adequately, which were identified by γH2AX. Foci of γH2AX and RAD51 in a minimum of 150 cells were counted automatized to determine the activity of HR. The red-dashed line showed the number of foci in the untreated control.

We observed an increase in the number of RAD51 foci in the healthy-donor skin fibroblasts (SBLF7, SBLF9) and melanoma cell lines ANST and LIWE following CC-115 treatment and radiation treatment compared to the untreated control (red-dashed line) and thus these cell cultures/lines were defined as HR-proficient (Figure 5). In contrast, we detected a reduction of RAD51 foci in the melanoma cell lines ICNI, RERO, ARPA, HV18MK, A375M, Mel624, and PMelL, which were defined as HR-deficient.

## 3. Discussion

Targeting the DNA-damage response (DDR) in tumor cells has become an important strategy in cancer treatment [64,65]. Many DDR-targeted drugs affect DSB repair by inhibition of HR or cell cycle checkpoints [66]. Approximately 50% of all cancer patients receive radiotherapy [9,10] which is known to cause DSBs [7,8]. Malignant cells often lack a proficient HR [67,68,69]. Targeting NHEJ as an alternative to HR regarding DSB repair extends the strategy of DDR-targeted therapies. CC-115 not only targets NHEJ by DNA-PK inhibition but also inhibits mTOR [34,35,36] and is currently studied in phase I (NCT01353625, NCT02833883) and phase II (NCT02977780) clinical trials. These trials study the treatment with CC-115 alone as well as the combination with radiotherapy or with enzalutamide, a nonsteroidal antiandrogen. The recently published results of the first-in-human phase I study of CC-115 showed peak plasma concentrations (C_max_) of 0.6 (± 0.1) µmol/L at the maximum tolerated dose [37]. These recent outcomes show that concentrations of CC-115 in our experiments, especially regarding colony forming assay, were close to concentrations obtainable in patients and therefore are of physiological relevance. Furthermore, the concentrations in our experiments are in line with concentrations used in previous pre-clinical studies [34]. Radiation as an important component of cancer therapy is mostly performed fractionated with single doses of 2 Gy which is why we irradiated cells at least with this dose. Regarding the full treatment scheme, total doses of 50-60 Gy are common in vivo. Thus, observable effects in our experiments following a single dose irradiation with 2 Gy that are significant but rather small could accumulate during fractionated treatment and therefore increase in a clinical relevant matter [70,71,72,73].

In general, cells are most sensitive to IR during mitosis and G2 phase [56,74]. A decrease of G2/M phase under combined inhibitor and IR treatment was observed only in PMelL that also showed a small increase in cell death when treated and a DEF of 1.0. However, none of the cell lines whose cell death rates were moderately or strongly increased by combined CC-115 and IR treatment nor cell lines showing DEFs > 1 showed an increased number of cells in G2/M phase. Interestingly, we found less cells in G2/M phase in melanoma cell lines HV18MK and Mel624 which showed high/highest cell death rates under combined CC-115 and IR treatment. CC-115-treated malignant cells were partly affected very strong, even if in less IR sensitive cell cycle phases. In our experiments, the proportions of cells in G1 phase increased following CC-115 treatment in 4 melanoma cell cultures in which we therefore suggest a CC-115 induced G1 phase arrest. DNA damage evidently causes G1 phase arrest [75]. Furthermore, these data could possibly be explained by the inhibition of mTOR which is also known to induce G1 phase arrest [76]. Homologous recombination (HR) is a very precise DSB repair pathway mainly occurring in S and G2 phase when sister-templates are available [13]. Thus, an induced G1 phase arrest leads to reduced HR activity. It also has been reported previously that the inhibition of DNA-PK activity in general and CC-115 in particular decreases HR in cells [35,77,78,79]. Non homologous end joining (NHEJ) as alternative DSB repair pathway is dominant in G0/G1 and G2 phase [80] but depends on functioning DNA-PK [18,81,82] which is inhibited by CC-115 [34,36,83]. Published data indicate a key role of NHEJ in radio- and chemo-resistance and that DNA-PK deficiency in cells leads to sensitization to DNA damaging agents [84]. Therefore, blocking CC-115 could lead to controversial effects.

To assess the functionality of HR we performed an HR assay. The NHEJ pathway was blocked by inhibiting DNA-PK, a key protein in this pathway. Cell lines were defined as HR-proficient when an increase of RAD51 was observed in comparison to an untreated control. Otherwise, cell lines were defined as HR-deficient when an inefficient HR pathway led to less RAD51 foci. The healthy-donor skin fibroblasts (SBLF7, SBLF9) were shown to be HR-proficient as well as the melanoma cell lines ANST and LIWE. The remaining seven out of nine melanoma cell lines exhibited a HR deficiency. Our findings about the HR status of our analyzed cells could not explain the differences in the overall response. DNA damage, if not repaired, is a powerful activator of cell death such as apoptosis and necrosis [17,85,86]. As a result of NHEJ inhibition and decreased HR/HR deficiency, we suggest that less DSBs are repaired properly leading to more apoptotic and necrotic cells, correlating with higher cell death rates, especially after DSB inducing events like radiotherapy [7,8]. The focus of the present study was to assess if simultaneous therapy with CC-115 and IR is superior in terms of tumor control to sequential treatment with, e.g., pausing targeted therapy during IR. In all colony forming assays and most of the experiments investigating apoptosis and necrosis the combined IR and inhibitor treatment compared to IR treatment alone lead to lower survival fractions or higher cell death rates, which supports this suggestion. Noticeably, IR alone showed relatively low cell death rates in our FACS analysis, whereas IR alone lead to significant reduction of cell survival in colony forming assays. Because IR is known to induce DNA damage like single and double strand breaks and therefore not just cell death but senescence too, it is plausible that effects are increased in the colony forming assay. These damages need time to accumulate during cell division to develop their full impact on cell survival. The combination of cell death analysis and colony forming assay, regarding overall a wide range of cellular mechanisms, helped us to understand biological coherences. Based on our findings, we suggest that CC-115 has radiosensitizing potential on certain melanoma but not on skin fibroblast cells. HR proficiency might explain findings in healthy-donor skin fibroblasts because DSBs are being repaired properly even when NHEJ is blocked by CC-115. Interestingly, we also observed inter-individual differences between melanoma cells, as some of the melanoma cell lines were affected much stronger by CC-115 than others. Other studies found that overexpression of ABCG2, a member of ATP-binding cassette transporters, results in increased CC-115 resistance [87]. The diversity of our cell lines represents the manifold range of genetic characteristics or mutations of tumors in cancer patients. Our in vitro data gives hint about the effects of combining irradiation with kinase inhibitor treatment, which should lead to the understanding that patients react quit different, between low and strong, and that it will be necessary to monitor them closely.

We also observed strong cytotoxic effects of CC-115 treatment alone and the HR functionality only correlated partially with the sensitivity of cells to inhibitor treatment, suggesting that the DSB repair inhibition is not the only cause for the effects we observed. It has been reported earlier that mTOR inhibition can induce apoptosis [88] and that combined mTOR and DNA-PK inhibition potently induces p53-independent cell death [34]. Inhibition of mTORC1 seems to affect cell proliferation but not cell survival [89] whereas mTORC2 inhibition appears to cause cell death induction in some cell lines [90]. The mTORC1 and mTORC2 proteins are both inhibited by CC-115 and may contribute to the effects we observed regarding cell death induction [35,36]. Regarding these two different target pathways of CC-115 the next step will be to investigate individual mTOR and DNA-PK inhibitors to differentiate the effects of CC-115.

## 4. Materials and Methods

### 4.1. Cell Culture

SBLF7 and SBLF9, both human skin fibroblasts, were extracted from healthy donors. Mel624, PMelL, and A375M are commercially obtained melanoma cell lines. HV18MK, ARPA, LIWE, ICNI, ANST, and RERO are patient derived melanoma cells, maintained by the department of Dermatology, University Hospital Erlangen [91] (Ethic approval No. 204_17 Bc). Primary human skin fibroblasts SBLF7 and SBLF9 were isolated from healthy donors via skin biopsy of the cutis and subcutis as described previously [92]. The biopsies were dissected and placed in tissue culture flasks. Each piece was covered with one drop of F-12 medium (Gibco, Waltham, MA, USA) supplemented with 40% fetal bovine serum (FBS) (Merck, Darmstadt, Germany). After attachment to culture flasks and outgrowth of the first fibroblasts, they were covered with medium that consisted of F-12 (Gibco, Waltham, MA, USA), supplemented with 12% FBS (Merck, Darmstadt, Germany), 2% non-essential amino acids (NEA) (Merck, Darmstadt, Germany), and 1% penicillin/streptomycin (Gibco, Waltham, MA, USA). After a series of cell divisions, fibroblasts were detached with trypsin and further cultured in medium as mentioned above, except FBS was enhanced to 15%. Melanoma cells were cultured in RPMI-1640 (Sigma Aldrich, München, Germany), supplemented with 20% FBS (Merck, Darmstadt, Germany), 1% L-Glutamin (Merck, Darmstadt, Germany), 1% hydroxyethyl-piperazineethane-sulfonic acid buffer (Merck, Darmstadt, Germany), 1% pyruvat-solution (Gibco, Waltham, MA, USA), 1% NEA (Merck, Darmstadt, Germany) and 0.04% gentamycin (Merck, Darmstadt, Germany). For experiments, the concentration of FBS was reduced to 2% in both media. Cell cultures did not exceed 40 passages. All cells were cultured in a humidified atmosphere at 37 °C in a 5% CO_2_ incubator.

### 4.2. Irradiation and Inhibitor

The cells were irradiated from 1 up to 8 Gy with ionizing radiation (IR) by an ISOVOLT Titan X-ray generator (GE, Ahrensburg, Germany). Unirradiated and irradiated samples were processed along each other. CC-115 (C16H16N8O, *M*_W_ 336.35 g/mol) was obtained from Selleck Chemicals (Houston, TX, USA), diluted in dimethyl sulfoxide (Roth, Karlsruhe, Germany) and stored at −80 °C. Prior to each experiment the required amount of CC-115 was freshly thawed.

### 4.3. Flow Cytometry

Cells cultured in flasks were washed with phosphate-buffered saline (PBS) (Sigma Aldrich, St. Louis, MO, USA) and detached using trypsin/EDTA (Gibco, Waltham, MA, USA). Thus, a single cell suspension was prepared. A suitable number of cells were seeded and incubated for 48 to 72 h so that 50–80% of the bottom of the flask was covered. For the prevention of an artificial increase of possible effects of treatment through stimulation of cell proliferation and for reduction of analytical interference [93,94] FBS-reduced cell culture medium (2% FBS), which included different concentrations of CC-115 (0.1–25 µmol/L) was used. Half of the treated and untreated cells were irradiated with 2 Gy IR 3 h after treatment. After an additional 48 h, cells were harvested by trypsination including the supernatant.

#### 4.3.1. Apoptosis and Necrosis Analysis

To detect apoptotic and necrotic cells using flow cytometry (Cytoflex, Beckman Coulter, Brea, CA, USA), harvested cells were suspended in Ringer solution and stained for 30 min on ice with Annexin V APC (BD, Heidelberg, Germany) and 7-amino-actinomycin D (7-AAD) (BD, Heidelberg, Germany). Kaluza Flow Cytometry Analysis 2.1 (Beckmann Coulter, Krefeld, Germany) was used to analyze the resulting data. Unstained cells (Annexin-negative/7-AAD-negative) were defined as viable cells. Annexin-positive/7-AAD-negative cells were defined as apoptotic and Annexin-positive/7-AAD-positive cells as necrotic (Figure 2A).

#### 4.3.2. Cell Cycle Analysis

For cell cycle analysis, harvested cells were fixed in 1 mL of FBS-reduced cell culture medium and 10 mL of 70% ethanol (Roth, Karlsruhe, Germany) for at least 12 h at 4 °C. Thereafter, the cells were suspended in Ringer solution and stained for 60 min on ice with Hoechst 333258 (Invitrogen, Eugene, OR, USA) and examined by flow cytometry (Cytoflex, Beckman Coulter, Brea, CA, USA). The resulting data were analyzed by Kaluza Flow Cytometry Analysis 2.1 (Beckmann Coulter, Krefeld, Germany) showing the distribution of cells in the G0/G1, S and G2/M phases of the cell cycle.

### 4.4. Colony Forming Assay

A preassigned number of cells was seeded in 60 mm petri dishes (Thermo Fisher Scientific, Roskilde, Denmark) containing 5 mL of fresh medium. After 4 h, different concentrations of CC-115 were added to the treatment group. After another 3 h, cells were irradiated with 1 to 5 Gy. 48 h post-irradiation cultures were rinsed, and fresh, drug-free medium was added. After 14 to 17 days cells were stained with methylene blue (#66725, Sigma Aldrich, München, Germany) for 30 min and thereafter colonies containing > 50 cells were counted. Plating efficiency (PE) was calculated as the ratio of number of colonies formed to the number of cells seeded. Survival fraction (SF) was calculated as the number of colonies formed, divided by the PE multiplied by the number of cells seeded [95]. Irradiation survival curves for untreated and treated (1 µmol/L CC-115) cells were plotted and an additional radiation survival curve was generated after normalizing for the cytotoxicity induced by CC-115. The dose enhancement factor (DEF) of different cell lines was calculated as a ratio of the estimated radiation dose at a certain SF of untreated cells to the normalized CC-115-treated cells [24,59,60,61].

### 4.5. Homologous Recombination Assay (RAD51 Immunofluorescence Staining)

The cells were seeded on cover slides. After reaching 90% confluence the medium was exchanged, cells were treated (5 µmmol/L of kinase inhibitor CC-115), and after 24 h of incubation irradiated additionally with a 10 Gy dose. The cells were fixed and permeabilized after 4 h (with 4% formaldehyde and 0.1% Triton ×-100/PBS) for 15 min at room temperature. After overnight blocking (1% bovine serum albumin; SERVA Electrophoresis GmbH, Heidelberg, Germany) slides were stained with primary antibodies mouse anti-γH2AX (1:1500, Merck, Darmstadt, Germany) and rabbit anti-Rad51 (1:250, abcam, Cambridge, UK) and secondary antibodies AlexaFluor488 goat anti-mouse and AlexaFluor594 chicken anti-rabbit (Invitrogen, Eugene, OR, USA) at +8 °C. [63]. Cell DNA was stained using DAPI (Sigma Aldrich, St. Louis, MO, USA) and cover slides were transferred onto glass slides using Vectashield (Vector Laboratories, Burlingame, CA, USA). Image acquisition was performed by a Zeiss Axio Plan 2 fluorescence microscope (Zeiss, Göttingen, Germany). Automated quantification was done by use of Biomas Software (MSAB, Erlangen, Germany).

### 4.6. Statistics

IBM SPSS Statistics 24 software was used to perform statistical analysis. All (statistical analyzed) experiments were performed with *n* ≥ 3, as declared precisely in the respective figure description. One/two-tailed Mann-Whitney-U test was used to analyze data. *p*-value ≤ 0.05 was determined as significant. Data are presented as mean ± standard deviation (SD). Graphs were generated using GraphPad Prism 8 software.

### 4.7. Biosecurity and Institutional Safety Procedures

Standard biosecurity and institutional safety procedures were followed.

### 4.8. Ethics Approval and Consent to Participate

Ethical approval was obtained in the Department of Dermatology, Universitätsklinikum Erlangen following approval by the institutional review board (Ethik-Kommission der Friedrich-Alexander-Universität Erlangen-Nürnberg, approval No. 204_17 Bc). The patients provided written informed consent.

## 5. Conclusions

In conclusion, we demonstrated that the dual inhibition of mTOR and DNA-PK by CC-115 led to cytotoxic effects of varying degrees in most of the melanoma cell lines. Dose enhancement factors (DEF) were calculated for each cell line as ratio of the radiation dose at SF 0.7 of untreated cells to the normalized CC-115-treated cells. DEF > 1 indicates that the inhibitor treatment induced radiosensitivity [23,58,59,60]. Normalized CC-115 treatment showed clearly increased effects for a combination of inhibitor and IR in 5 out of 7 cancer cell lines. Healthy cells were hardly affected or not at all. CC-115 showed radiosensitizing potential only in melanoma cells. Phase I studies revealed that CC-115 is tolerable in vivo and is able to overcome the blood–brain barrier. Our results support to study the combination of CC-115 and radiotherapy in clinical trial. CC-115 treatment could be a promising approach for patients with metastatic melanoma particularly in the combination with radiotherapy.

## Figures and Tables

**Figure 1 ijms-21-09321-f001:**
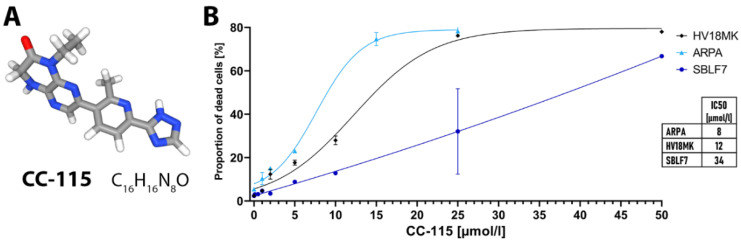
Structural formula of CC-115 and dose escalation study. (**A**) Chemical structural formula of CC-115 (C16H16N8O) [54] (**B**) Healthy donor skin fibroblasts (SBLF7) and patient derived melanoma cells (ARPA, HV18MK) were treated for 48 h in a dose escalation study with 0.1–25 µmol/L CC-115. Dead cells were defined as sum of apoptotic and necrotic cells measured using flow cytometry (apoptosis/necrosis analysis by Annexin V-APC/7-AAD staining). Half maximal inhibitory concentrations (IC50) for the different cell line are displayed. Each value represents mean ± SD.

**Figure 2 ijms-21-09321-f002:**
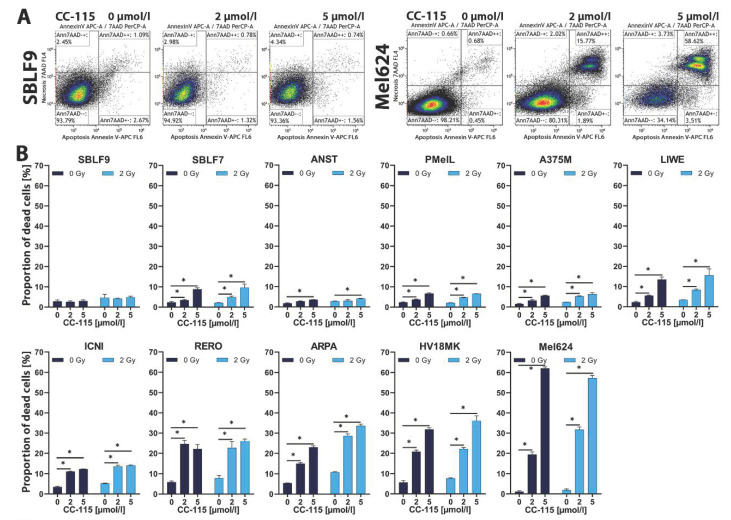
Cell death induced by CC-115 treatment with/without IR. (**A**) Flow cytometry was used for apoptosis/necrosis analysis by V-APC/7-AAD staining. Unstained cells (Annexin-negative/7-AAD-negative) were defined as viable cells. Annexin-positive/7-AAD-negative cells were defined as apoptotic and Annexin-positive/7-AAD-positive cells as necrotic. Examples for gating at different CC-115 concentrations (0, 2 and 5 µmol/L) are shown, left: healthy-donor skin fibroblasts: SBLF9, right: melanoma cells: Mel624. (**B**) Graphs show proportion of dead cells (defined as sum of apoptotic and necrotic cells) at different CC-115 concentrations (0, 2 and 5 µmol/L) with/without IR (2 Gy) in skin fibroblast (SBLF9, SBLF7) and melanoma cells (RERO, ANST, A375M, LIWE, ICNI, PMelL, ARPA, HV18MK, Mel624). Each value represents mean ± SD (*n* = 3). Significance was determined by one-tailed Mann-Whitney U test * *p* ≤ 0.05.

**Figure 3 ijms-21-09321-f003:**
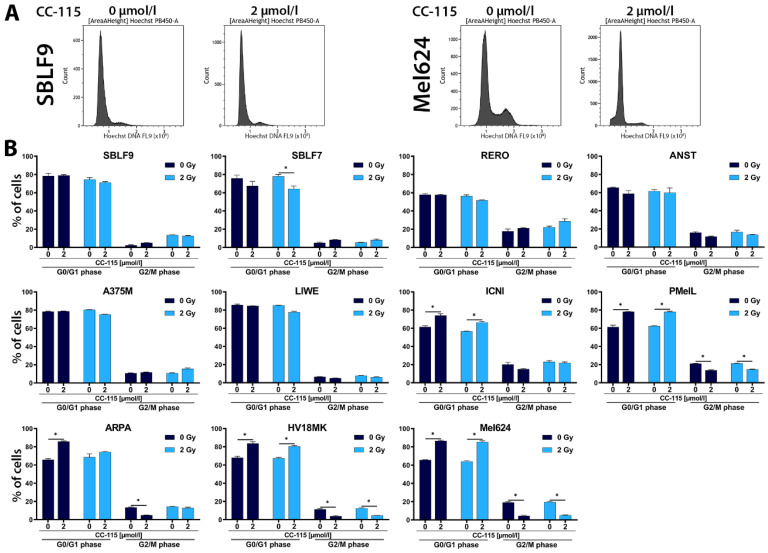
Cell cycle distribution under CC-115 treatment with/without IR. (**A**) Flow cytometry was used for cell cycle analysis by Hoechst staining. Examples of representative histograms of Hoechst stained DNA distribution for control and 2 µmol/L CC-115 are shown, left: healthy-donor skin fibroblasts (SBLF9), right: melanoma cells (Mel624). (**B**) Graphs show the percentage of cells in G0/G1 phase (left columns) or G2/M phase (right columns) for 0 and 2 µmol/L CC-115 with/without IR (2 Gy) in skin fibroblasts (SBLF9, SBLF7) and melanoma cells (RERO, ANST, A375M, LIWE, ICNI, PMelL, ARPA, HV18MK, Mel624). Each value represents mean ± SD (*n* = 3). Significance was determined by one-tailed Mann-Whitney U test * *p* ≤ 0.05.

**Figure 4 ijms-21-09321-f004:**
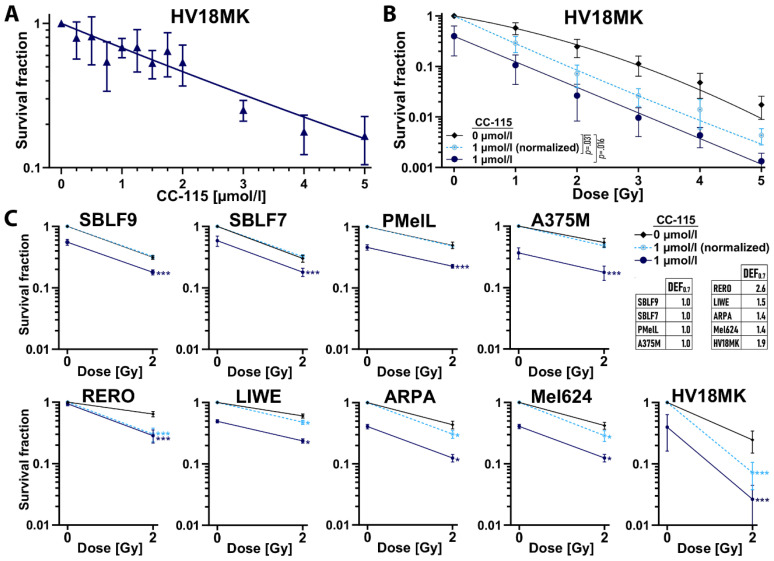
Colony forming assay to study the radiosensitizing potential of CC-115. (**A**) Decreasing survival fractions (SF) in colony forming assay under increasing CC-115 concentrations (0.25–5 µmol/L) in melanoma cells culture HV18MK. (**B**) Colony forming assay of HV18MK cells treated with up to 5 Gy IR with/without 1 µmol/L CC-115. Each value represents mean ± SD (*n* = 4). Solid lines (black, blue) represent mean SF. Dashed line (turquoise) represents mean SF normalized to the cytotoxicity induced by CC-115 alone. Significance was determined by one-tailed Mann-Whitney U test; *p* = 0.031 comparing control to normalized CC-115 treatment; *p* = 0.016 comparing control to CC-115 treatment. (**C**) Colony forming assay of healthy-donor skin fibroblasts (SBLF9, SBLF7) and melanoma cells (PMelL, A375M, RERO, LIWE, ARPA, Mel624, HV18MK) treated with/without 2 Gy IR with/without 1 µmol/L CC-115. Solid lines (black, blue) represent mean SF. Dashed line (turquoise) represents mean SF normalized to the cytotoxicity induced by CC-115 alone. Significance was determined by two-tailed Mann–Whitney U test comparing control to normalized CC-115 treatment (turquoise asterisks) or control to CC-115 treatment (blue asterisks), * *p* ≤ 0.05, *** *p* ≤ 0.001 (*n* = 4). Mean dose enhancement factors (DEF0.7) are shown and were calculated as ratio of the estimated radiation dose at survival of 70% of untreated cells to the normalized CC-115-treated cells.

**Figure 5 ijms-21-09321-f005:**
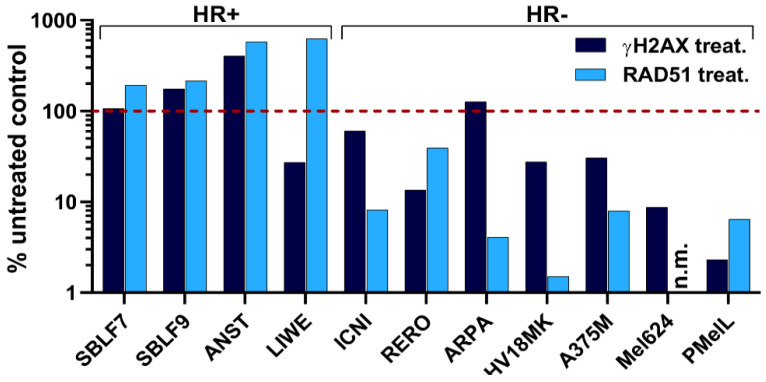
Homologous recombination status based on RAD51 foci in untreated and treated cells. Homologous recombination functionality was evaluated by measuring DNA double-strand breaks after 24 h of exposure to DNA-PK inhibitor CC-115, followed by irradiation with a 10 Gy dose. Irradiation-induced DSBs were analyzed via γH2AX foci 4 h after irradiation. Red-dashed line shows foci number in untreated control. Increase or decrease of RAD51 foci number under CC-115 treatment indicated HR-proficiency or HR-deficiency. SBLF7, SBLF9, ANST and LIWE increase their number of RAD51 foci in the treated sample compared to the control (HR-proficient). Melanoma cells ICNI, RERO, APRA, HV18MK, A375M, Mel624 and pMelL showed less RAD51 foci after treatment with the DNA-PK inhibitor (HR-deficient). (n.m. = not measurable)

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
