# Peer review of "Dual mTOR/DNA-PK Inhibitor CC-115 Induces Cell Death in Melanoma Cells and Has Radiosensitizing Potential"

_ijms, 2020, doi:10.3390/ijms21239321_

Round 1

Reviewer 1 Report

The manuscript of Felix Bürkel et al describes the effects of CC-115, which is CC-115 is a dual mTORC1/mTORC2 and DNA-PK inhibitor, on the viability of melanoma cells in comparison to healthy-donor skin fibroblasts. The authors used for their experiments two skin fibroblast cell cultures of healthy donors, three melanoma cell lines and six patient-derived melanoma cell cultures. To evaluate the effects of CC-115, apoptosis/necrosis and cell cycle analyses as well as colony forming assay were carried out, and they were supplemented by the immunofluorescence microscopy assay to determine the homologous recombination activity in the treated cells.

Major critic:

Basing on their results, the authors made a conclusion in the abstract section that “CC-115 showed radiosensitizing potential only in melanoma, but not in healthy skin fibroblasts.” However, the conclusion is not entirely accurate, since the degree of increase in cell mortality upon treatment with the inhibitor strongly depends on the cell line both for malignant and non-malignant cell lines. Indeed, the inhibitor doesn’t influence the viability of healthy-donor skin fibroblasts SBLF9 and melanoma cells ANST and produces a comparable low effect on healthy-donor skin fibroblasts SBLF7 and melanoma cells PMelL and A375M (see fig. 2). Analysis of the homologous recombination status based on RAD51 foci in untreated and treated cells shows an increase in HR activity not only in both healthy-donor skin fibroblast cell lines but also in melanoma cell lines ANST and LIWE. Unfortunately, the authors don’t discuss in detail the reasons for such different behavior of different cell lines in response to treatment with CC-115, and don’t propose additional experiments to understand this difference. As a result, the scientific significance of the work is significantly reduced.

Minor critics:

Line 137: “Effects of IR alone compared to control tended to be rather small or non-existent” – It’s a rather strange remark since figure 4 shows a significant decrease of survival cell fraction under irradiation alone for all the cell lines. And it is strange that tr IR effect is only noticeable in Fig. 4, whereas in Fig. 2 and 3 it is really practically not noticeable.

Fig. 4: CC-115 has no effect on the viability of healthy-donor skin fibroblasts SBLF9 ion fig. 2, whereas it has this effect on fig. 4. How can the authors explain this?

Line 214: the authors measured HR activity of cells considering that “when blocking the second major DDR pathway NHEJ, the cells should be forced to use HR”. This statement is true but not completely. It is known that in addition to the canonical NHEJ pathway, cells maintain an alternative end-joining pathway that engages various factors, such as the MRN complex, PARP-1, WRN, and LIG1 (Symington LS, Gautier J. Annu Rev Genet. 2011; 45:247–271). Obviously, it's worth checking how NHEJ is really suppressed in different cell lines upon CC-115 treatment. Perhaps that would explain the difference in their response to the inhibitor treatment noted above  

It is necessary to carefully check the grammar, as there are many inaccuracies and errors in the article.

In particular:

Line 23: the phrase is not clear.

Line 57: the verb form “is presently been conducted” is not correct.

Line 83: “systematic therapies” should be replaced by “systemic therapies”

Line 89: It is better to remove the word “therapeutic “ in the expression “undesirable therapeutic effects”

Reviewer 2 Report

This is an interesting article with important new findings. The study is good designed. The theoretical part is sufficiently described and it is clear from it what the authors wanted to investigate in the study.

It is well done and the methods and results are clearly described. The discussion and references are adequate.

I have no specific objections about the article but I want to suggest that Author should reduce the number of reference here.

Author Response

Dear Reviewer 2,

thank you for giving us the opportunity to submit a revised draft of my manuscript titled “Dual mTOR/DNA-PK
Inhibitor CC-115 induces Cell Death in Melanoma Cells and has Radiosensitizing Potential” to “International
Journal of Molecular Science – Molecular Oncology”. We appreciate the time and effort that you and the
reviewers have dedicated to providing your valuable feedback on our manuscript. We are grateful to the
reviewers for their insightful comments on our paper. We have been able to incorporate changes to reflect
the suggestions provided by the reviewers. We have highlighted the changes within the manuscript.

To reviewer #2:
This is an interesting article with important new findings. The study is good designed. The theoretical part is
sufficiently described and it is clear from it what the authors wanted to investigate in the study.

It is well done and the methods and results are clearly described. The discussion and references are
adequate.

I have no specific objections about the article but I want to suggest that Author should reduce the number of
reference here.

Thank you very much for your feedback. We appreciate your response and focus on a smaller number
of references more consequently.

Round 2

Reviewer 1 Report

The manuscript may be accepted in the present form